# Abundance, Biomass, and Production of Bacterioplankton at the End of the Growing Season in the Western Laptev Sea: Impact of Khatanga River Discharge (Arctic)

Alexander I. Kopylov [1,*], Dmitriy B. Kosolapov [1], Anna V. Romanenko [1], Elena A. Zabotkina [1] and Andrey F. Sazhin [2,*]

1 Papanin Institute for Biology of Inland Waters, Russian Academy of Sciences, 109, 152742 Borok, Russia; dkos@ibiw.ru (D.B.K.); romanenko68@list.ru (A.V.R.); zabel@ibiw.ru (E.A.Z.)
2 Shirshov Institute of Oceanology, Russian Academy of Sciences, 36, Nakhimovskiy Prospekt, 117997 Moscow, Russia
* Correspondence: kopylov@ibiw.ru (A.I.K.); andreysazhin@yandex.ru (A.F.S.)

**Abstract:** The structure and productivity of planktonic microbial communities in the ecosystems of the Siberian Arctic seas are significantly dependent on freshwater input. During the study, we determined the spatial distribution of the abundance, biomass, and production of heterotrophic bacterioplankton in the Western Laptev Sea on the transect from the Khatanga River estuary to the continental slope and assessed the impact of river freshwater discharge. The influence of fresh water on bacterioplankton was restricted mainly to Khatanga Bay (KHAB) and the transitional zone (TZ) and was poorly recognized in the Western shelf (WS) and continental slope (CS) areas. The total bacterial abundance decreased from KHAB to the CS. Particle-attached bacteria constituted on average 63.0% of the total abundance of bacterioplankton in KHAB and 1.0% at the CS. Average bacterial production in the water column was highest in KHAB (10.3 mg C m$^{-3}$ d$^{-1}$), decreasing towards the CS (0.7 mg C m$^{-3}$ d$^{-1}$). In KHAB and TZ, bacteria were the main component of the planktonic community (44−55%). These results show that at the end of the growing season, bacterial processes prevailed over autotrophic ones and contributed largely to the total biological carbon flux in the coastal ecosystem of the Western Laptev Sea.

**Keywords:** Arctic; Laptev Sea; bacterioplankton distribution; particle-attached bacteria; bacterial growth efficiency; bacterial production

## 1. Introduction

The Arctic Ocean is strongly influenced by inflow from large rivers, and the discharge of these inputs is increasing [1], with implications for the delivery of dissolved and particulate organic matter to inshore arctic seas [2]. The Arctic Ocean receives near 11% of global river runoff, while its volume is only 1% of the world's ocean's volume [3,4]. In estuarine and coastal zones, fresh water and terrestrial material strongly influence the distribution and productivity of autotrophic and heterotrophic organisms [5]. Heterotrophic bacteria are the major consumers of dissolved organic matter (DOM) and are an important component of planktonic communities [6–8]. DOM provides energy and carbon to bacteria, and bacteria function as a sink (mineralization of DOM to $CO_2$) or link (generation of transportable biomass, which enters higher levels of the food web or as a result of viral lysis), where the carbon of lysed cells does not pass to higher trophic levels but enters the environment ("viral shunt") and is actively involved in the metabolism of heterotrophic bacteria [9–11].

Coastal marine systems in the Arctic typically contain a high concentration of suspended organic particles, which enter the water column via river runoff, glacier/ice sheet melting, and permafrost melting [12,13]. Bacteria are important decomposers and deminer­alizers of particulate organic matter (POM) [14]. Where particulate material is abundant

(estuarine environments), attached bacteria can make up a significant proportion of the total bacterial abundance [15]. A global review by Simon et al. [16] indicated that particle-attached bacterial production may exceed 30% of total bacterial production in riverine and estuarine systems, but in pelagic oligotrophic and mesotrophic environments, it generally accounts for less than 14%.

In the Beaufort Sea (Canadian Arctic), studies have highlighted the importance of bacteria degrading POM in aggregates [17,18] and have reported a high variability (from 0 up to 98%) of the contribution of particle-attached bacteria to total bacterial production [19].

At the end of the 20th century, researchers [20] suggested that bacterial activity is strongly suppressed by low water temperatures in the polar region. Early 21st century studies in Arctic waters demonstrated the high metabolic activity of bacteria [11,21–23] even at near- or subzero temperatures. This data suggest that low concentrations of DOM and high mortality from viral lysis and bacterivory limit bacterial production rather than low temperatures in seawater [24–27].

The largest continental shelf areas of the world are located in the Siberian Arctic; they are impacted by large volumes of fresh water from Siberian river basins, which transport a high amount of organic matter and nutrients into the Arctic Ocean. The Laptev Sea annually receives 820 km$^3$ of river runoff, the second-highest (after the Kara Sea) freshwater contribution to the Arctic seas [28,29]. The Khatanga River is the second-largest river flowing into the Laptev Sea, with an annual runoff of 105 km$^3$. The suspended particulate matter (SPM) concentration in the Khatanga River reaches $140-170$ g m$^{-3}$. The freshwater runoff of the Khatanga River is small; despite this, the plume formed by this runoff has an anomalously large area, similar to that of the Ob and Yenisei plumes [30].

Previous studies of the spatial variability of the abundance of free-living bacteria and bacterial production in the Laptev Sea were carried out mainly in the zone of influence of Lena River runoff [31–34]. The contribution of particle-attached bacteria to the total bacterial abundance and biomass on the Laptev Sea shelf has not been assessed.

We hypothesized that freshwater impact would give rise to dissolved organic carbon (DOC) and particle loading, which in turn would affect the spatial distribution, composition, and productivity of bacterial communities on the Western shelf of the Laptev Sea. We tested this hypothesis by analyzing a transect from the inner part of Khatanga Bay to the continental slope.

The central aim of the present study was to determine the spatial distribution of the abundance, biomass, and production of heterotrophic bacterioplankton depending on variations in the physic and chemical conditions on the Western shelf of the Laptev Sea.

Our tasks in this study were to assess (1) the spatial distribution of POM (3–210 µm fraction) with attached bacteria, (2) the contribution of particle-attached bacteria to total bacterial abundance and biomass, (3) the spatial distribution of total bacterial abundance, biomass, and production and their relationship with changing environmental conditions along the transect, and (4) the role of bacterioplankton in forming the total biomass of the plankton community in different water masses of the Laptev Sea.

## 2. Materials and Methods

### 2.1. Study Sites and Sampling

Water samples were collected during cruise 69 of the R/V Akademik Mstislav Keldysh, during 17–20 September 2017, at eight sampling stations in the Western Laptev Sea (Figure 1, Table 1).

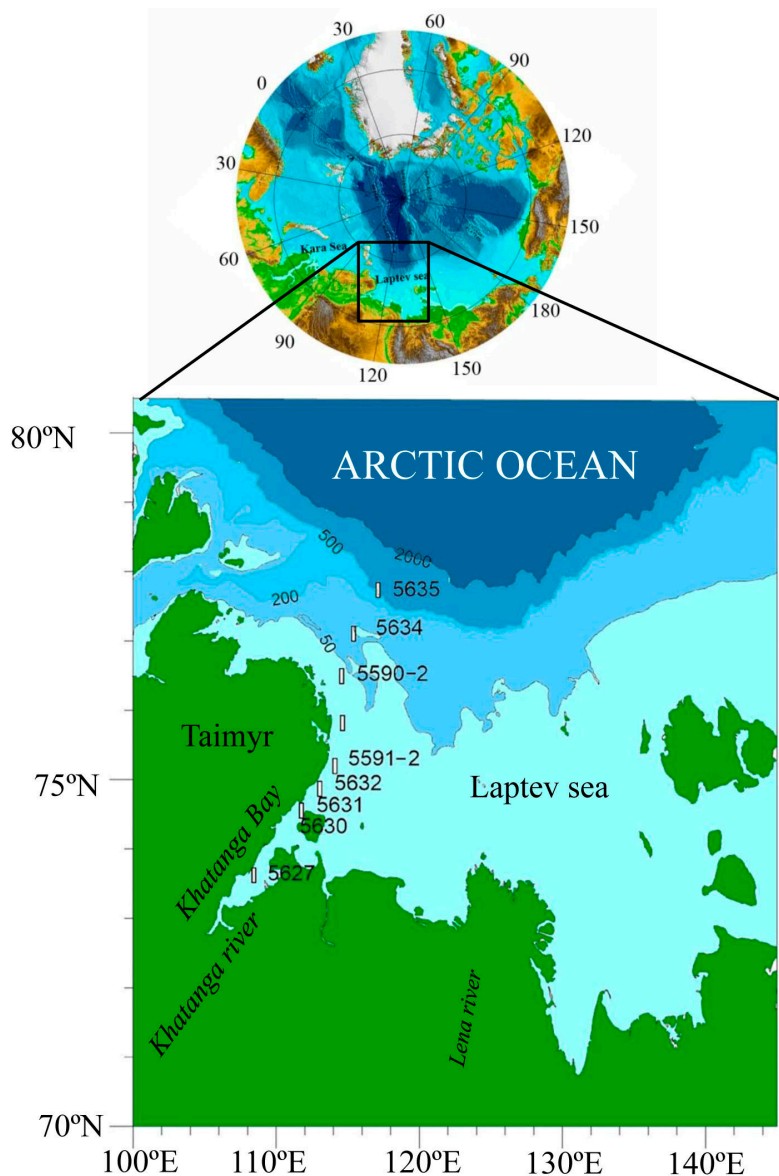

**Figure 1.** Sketch map of locations of sampling stations.

**Table 1.** Sampling details and environmental characteristics at stations in the Western Laptev Sea.

| Stn. Name | Latitude, N | Length, E | Depth, m | Alkalinity, mg eq L$^{-1}$ | $NO_2^-$ $NO_3^{-\,a}$, μM | $NH_4^{+\,a}$, μM | $PO_4^{3-\,a}$, μM | DOC [b], μM |
|---|---|---|---|---|---|---|---|---|
| | | | | Khatanga Bay | | | | |
| 5627 | 73°49′ | 108°18′ | 15 | 0.65 * | 1.10 | 2.72 | 0.20 | 650 |
| 5630 | 74°25′ | 110°34′ | 27 | 1.58 | 0.57 | 1.15 | 0.20 | 438 |
| | | | | Ttansition Zone | | | | |
| 5631 | 74°58′ | 111°67′ | 29 | 1.81 | 0.52 | 0.82 | 0.15 | 395 |
| 5632 | 74°84′ | 113°80′ | 33 | 1.99 | 0.74 | 0.55 | 0.16 | 387 |
| | | | | Western shelf | | | | |
| 5591-2 | 75°42′ | 115°41′ | 45 | 2.11 | 0.59 | 0.43 | 0.22 | 278 |
| 5590-2 | 77°17′ | 114°67′ | 64 | 2.25 | 1.05 | 0.27 | 0.25 | 209 |
| 5634 | 77°64′ | 115°53′ | 183 | 2.24 | 3.83 | 0.11 | 0.33 | 220 |
| | | | | Continental slope | | | | |
| 5635 | 78°05′ | 115°88′ | 860 | 2.30 | 7.47 | 0.10 | 0.44 | ND |

*—mean concentrations in water column; [a] Data from Sukhanova et al., 2019 [35]; [b] Data from Drozdova et al., 2021 [36]; $NO_2^-$ $NO_3^-$—concentrations of nitrite and nitrate [35]; $NH_4^+$—concentrations of ammonia [35]; $PO_4^{3-}$—concentrations of phosphate [35] and DOC—concentrations of dissolved organic carbon in the water column [36], ND—no data.

In the Western Laptev Sea, from the inner part of Khatanga Bay (depths up to 15 m) to the continental slope area in the north (depth 860 m), four biotopes were identified, characterized by different properties of the pelagic environment, composition, and quantitative characteristics of bacterioplankton: Khatanga Bay (stations 5627, 5630), the transition zone between Khatanga Bay and the shallow shelf of the Laptev Sea (stations 5631, 5632), the Western shelf (stations 5591-2, 5590-2, 5634), and the continental slope (station 5635). Water samples were collected with 5-L Niskin bottles on a Rosette 32 sampler equipped with a CTD (SBE-911), (Sea-Bird Scientific, Bellevue, WA, USA). Temperature and salinity data were obtained with the SBE-911 using temperature and conductivity sensors. Dissolved organic carbon was measured as described previously [36]. Water samples for microscopic studies were fixed immediately after sampling with 25% glutaraldehyde (final concentration 2%).

### 2.2. Enumeration of Bacteria and Suspended Particles

The abundances and sizes of bacteria and suspended particles were determined by epifluorescence microscopy and with fluorochrome 4′, 6′–diamidino–2–phenylindole (DAPI) [37]. This technique does not distinguish between archaea and bacteria; for convenience, we, like other authors [19], use the term bacteria when referring to DAPI count and archaea when referring specifically to prokaryotes of this domain. From each sample, 2–5 mL of water was stained with DAPI at a concentration of 1 μg mL$^{-1}$ (final concentration) and filtered onto a black Nuclepore filter (0.2-μm pore size). Observations and counts were made under an Olympus BX51 epifluorescent microscope (Olympus, Tokyo, Japan) using CellF Image Analysis Software at ×1000 magnification. On each filter, at least 400 free-living bacteria were counted, and the dimensions of at least 100 cells were measured.

After staining with DAPI, which binds nucleic acid, under fluorescence microscopy, the majority of suspended particles glow yellow, indicating that these yellow particles are organic and represent nano- and microsized detritus [38]. On DAPI-stained filters with epifluorescence microscopy, yellow organic particles were clearly distinguished from bacterial cells [38,39]. The number of particle-attached bacteria was estimated by two methods: direct counting of bacteria on particles and sonication. The abundance of attached bacteria was assessed in accordance with methods reported in the reference [40]. To estimate the abundance of attached bacteria, particles 3–210 μm in size were classified into four size classes, considering the equivalent spherical diameter (ESD; i.e., the diameter of a sphere with equivalent volume to nonspherical-shaped particles [41]: 3–10 μm (size class 1), >10−20 μm (size class 2), >20−50 μm (size class 3), and >50−210 μm (size class 4). On each filter, at least 200 particles 3–210 μm in size were counted, and at least 20 particles of each size class were chosen randomly and inspected for the number of attached bacteria. On each filter, at least 400 particle-attached bacteria were counted, and the dimensions of at least 100 cells were measured.

This method reveals the abundance of bacteria, at least of surface-exposed microorganisms. Due to technical constraints (background fluorescence, only two-dimensional visualization), the true abundance may be underestimated [40,42].

The abundance of particle-attached bacteria sampled in Khatanga Bay and the transition zone was assessed by sonication. We dislodged bacteria from particles by sonication after [43]. Water samples were filtered onto 3-μm isopore-size membrane filters (TSTP, 25 mm diameter, Millipore) to obtain the particle-attached microbial fraction. Filters were stored in 4.5 mL of reconstituted seawater (32% solution of dry natural sea salt in deionized water, autoclaved at 1 atm) and fixed with 25% glutaraldehyde (final concentration 2%). Tetrasodium pyrophosphate (final concentration 5 mM) was added to the samples. After 15-min incubation on ice, the mixture was sonicated (three cycles: 30 s, 200 J, 20 kHz, and 30-s intervals with manual shaking of the subsample). Afterwards, samples were used to determine the abundance of bacteria.

In 15 water samples collected in KHAB and the transition zone (TZ), the ratio of the abundance of bacteria obtained using ultrasound and counted on the visible surface of suspended particles with a size of 3–210 µm varied from 1.7 to 3.4, averaging 2.2 ± 0.11 (Table 2). Taking this into consideration, at stations 5591-2, 5590-2, 5634, and 5635, where the SPM concentration was much lower and the sonication method was not used, the abundance of particle-attached bacteria obtained by the direct method increased twofold.

**Table 2.** Abundance ($10^3$ cells mL$^{-1}$) of particle-attached bacteria received by method of directly counting bacteria on visible particle surface detritus (PABD) and method with sonication (PABS).

| Station | Horizon, m | PABD | PABS | PABS/PABD |
|---------|-----------|-------|--------|-----------|
| 5627 | 0 | 320.3 | 640.5 | 2.0 |
|  | 5 | 580.9 | 999.3 | 1.7 |
|  | 12 | 816.4 | 1497.0 | 1.8 |
| 5630 | 0 | 350.5 | 806.2 | 2.3 |
|  | 5 | 308.5 | 1049.0 | 3.4 |
|  | 20 | 381.6 | 763.3 | 2.0 |
|  | 23 | 478.2 | 1000.1 | 2.1 |
| 5631 | 0 | 136.6 | 382.6 | 2.8 |
|  | 10 | 155.4 | 310.8 | 2.0 |
|  | 18 | 172.8 | 380.1 | 2.2 |
|  | 25 | 134.3 | 309.0 | 2.3 |
| 5632 | 0 | 100.9 | 282.6 | 2.8 |
|  | 10 | 27.9 | 55.8 | 2.0 |
|  | 17 | 75.9 | 151.8 | 2.0 |
|  | 22 | 47.7 | 100.2 | 2.1 |

The specific gravity of suspended particles was taken to be equal to 1. The wet biomass of bacteria was estimated from the individual cell volume using Image Scope Color image analysis software. The carbon content in bacterial cells (C, fg C cells$^{-1}$) was calculated using the following allometric equation: $C = 120 \times V^{0.72}$, where V is the mean volume of bacterial cells, µm$^3$ [44].

The number and biomass of heterotrophic nanoflagellates (HNF) were determined using fluorochrome Primulin and black Nuclepore filters (0.2 µm pore size) [45]. Observations and counts were made under an Olympus BX51 epifluorescent microscope (Olympus, Japan) using CellF Image Analysis Software at ×1000 magnification. The organic carbon content in the raw HNF biomass was taken at 22% [46].

### 2.3. Production of Bacterioplankton

The growth rate and production of heterotrophic bacteria were determined by dilution according to changes in their abundance in isolated water samples exposed to in situ temperatures in a temperature-controlled chamber for 36–40 h. The seawater samples were diluted 1:10 with water passed through a membrane filter (pore size 0.17 µm) to remove grazers of bacteria [47,48]. All experiments were conducted in triplicate. The specific growth rate of bacteria (m, h$^{-1}$) was calculated by the formula m = (lnN$_t$ − lnN$_o$)/t, where No and N$_t$ are the initial and final abundances of bacteria, respectively, and t is the incubation time, h. The production of bacteria (P$_{pr}$, mg C m$^{-3}$ day$^{-1}$) or cells (mL$^{-1}$ day$^{-1}$) was calculated by multiplying the specific growth rate by the biomass (or abundance) of bacteria in undiluted seawater. Bacterial carbon demand was calculated based on a gross efficiency of 27% [49].

### 2.4. Statistical Analyses

Correlations between the parameters were analyzed using Spearman's correlation coefficient calculated by Past 4.03 software [50] with regard to the prerequisites for the analyzed data.

## 3. Results

### 3.1. Environmental Conditions

Stations 5627−5630 in Khatanga Bay (KHAB), which is influenced by Khatanga River runoff, were characterized by relatively low salinity (3–17 psu) in the upper 5–10 m layer and relatively high water temperature (2.8–4.2 °C) (Figure 2). An anomalously high water turbidity was noted in the bay for the coastal regions of the Arctic seas: 51−80 NTU (nephelometric turbidity unit) m$^{-1}$ in the lower water layers and 42−44 NTU m$^{-1}$ in the upper [35]. The Secchi disk's water transparency was 0.5 m [51].

At stations 5631–5632 located in the TZ, in the upper 10 m water layer, salinity increased to 19–22 psu and water temperature decreased to 2.2–2.6 °C (Figure 2). At station 5631, turbidity decreased to 4.8 NTU m$^{-1}$, and at station 5632, it dropped to 1.5 NTU m$^{-1}$ [35].

At stations 5591-2, 5590-2, and 5634 located on the Western shelf (WS), the water temperature on the surface in the south–north direction decreased from 2.3 to −0.4 °C, while salinity, conversely, increased from 22.5 to 30.0 psu. At all stations, the water temperature near the bottom was negative: from −0.9 to −1.7 °C; salinity varied from 33.5 to 34.6 psu.

At station 5635, located in the continental slope (CS) area, the salinity in the water column changed insignificantly. At this station, the water temperature in the deep-water horizons was higher than that in the surface horizons (Figure 2). Warm saline (>34.6 psu) waters related to an admixture of Atlantic waters entering the Arctic Ocean [52,53] were detected at a depth of 140−850 m in the Northern part of the Khanatga transect. Water turbidity in the upper mixed layer decreased to 0.2 NTU m$^{-1}$, and water transparency increased to 15 m [35,51].

The waters of KHAB, which receives Khatanga River runoff, contained a large amount of DAPI-positive yellow suspended particles (DYP) [38]. The abundance of DYP 3–210 μm in size ($N_P$) in the KHAB water column varied within (19.3–282) × 10$^3$ particles mL$^{-1}$, averaging (139.0 ± 33.1) × 10$^3$ particles mL$^{-1}$ (Figure 2). The mass of DYP 3–210 μm in size ($M_P$) in the KHAB water column varied within 9.08–380.08 g m$^{-3}$, averaging 158.12 ± 46.1 g m$^{-3}$ (Figure 2).

In the TZ, $N_P$ decreased to (3.6–13.4) × 10$^3$ particles mL$^{-1}$, averaging (9.2 ± 1.2) × 10$^3$ particles mL$^{-1}$ (Figure 2). $M_P$ varied within 1.65–45.17 g m$^{-3}$, averaging 10.38 ± 4.8 g m$^{-3}$. DYP 50–210 μm in size were not found in the TZ.

On the WS, $N_P$ further decreased to (0.14–11.2) × 10$^3$ particles mL$^{-1}$, averaging (2.5 ± 0.8) × 10$^3$ particles mL$^{-1}$, and $M_P$, within 0.01–4.44 g m$^{-3}$, averaging 1.26 ± 0.46 g m$^{-3}$ (Figure 2). The average $N_P$ for the water column on the inner shelf was five times higher than on the outer.

In the area of CS, $N_P$, and $M_P$ were the lowest on the transect, respectively, from 0.03 to 0.22 × 10$^3$ particles mL$^{-1}$ (average, (0.14 ± 0.02) × 10$^3$ particles mL$^{-1}$) and from 0.01 to 0.1 g m$^{-3}$ (average, 0.05 ± 0.01 g m$^{-3}$). Thus, the average $N_P$ and $M_P$ values on the CS were lower than those in KHAB, respectively, by 1000 and 3000 times. DYP 20–50 and 50–210 μm in size were not detected at station 5635.

The abundance ($N_{PB}$) and mass ($M_{PB}$) of DYP 3–210 μm in size with attached bacteria significantly decreased with distance from KHAB (Table 3). In the studied biotopes, with the exception of the CS, the NPB/NP and MPB/MP ratios were the lowest in the 3–10 μm size class. DYP larger than 20 μm were always colonized by bacteria (Table 3). Among DYP with attached bacteria, particles 3–10 μm in size, as a rule, predominated in terms of abundance, and particles 10–20 or 20–50 μm in size dominated in terms of mass (Figure 3).

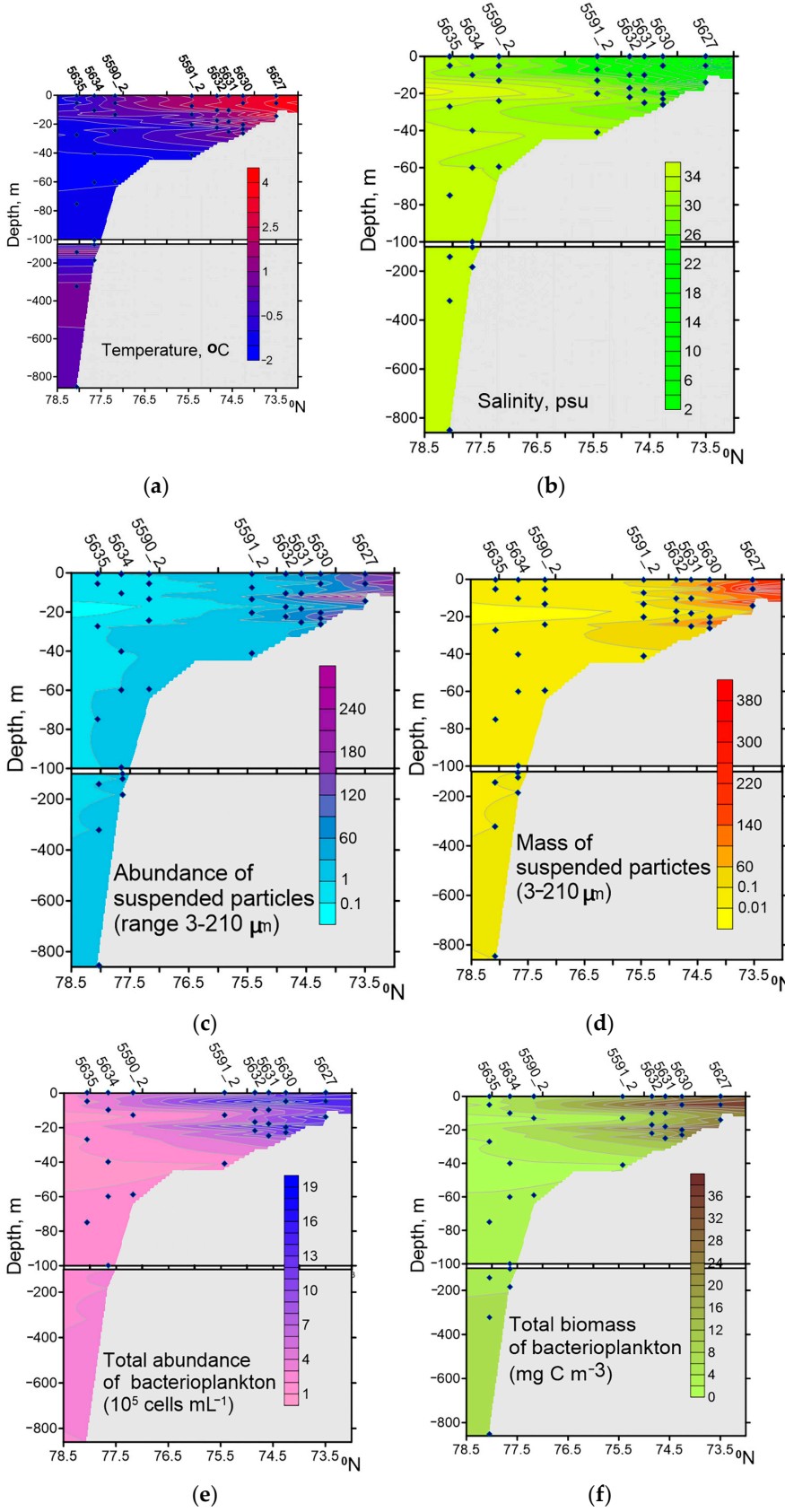

**Figure 2.** Distribution of (**a**) temperature, °C; (**b**) salinity, psu; (**c**) abundance of suspended particles (range 3–210 μm), $10^3$ particles mL$^{-1}$; (**d**) mass of suspended particles (range 3–210 μm), g m$^{-3}$; (**e**) total abundance of bacterioplankton, $10^5$ cells mL$^{-1}$; (**f**) total bacterioplankton biomass, mg C m$^{-3}$ along the transect.

**Table 3.** Mean (±SE) abundance ($N_{PB}$, $10^3$ particle $mL^{-1}$) and mass ($M_{PB}$, g $m^{-3}$) of suspended particles with attached bacteria of different size classes and their share in the total number ($N_{PB}/N_P$, %) and mass ($M_{PB}/M_P$, %) of suspended particles of the corresponding size class in different biotopes in the Western Laptev Sea.

| Parameters | Particle Size Classes, μm | | | | |
|---|---|---|---|---|---|
| | 3–10 | >10–20 | >20–50 | >50–210 | 3–210 |
| | Khatanga Bay | | | | |
| $N_{PB}$ | 44.7 ± 12.2 | 27.8 ± 8.7 | 18.4 ± 7.0 | 0.2 ± 0.03 | 91.1 ± 26.1 |
| $N_{PB}/N_P$, % | 50.2 ± 4.0 | 75.5 ± 3.9 | 100 | 100 | 63.1 ± 6.1 |
| $M_{PB}$ | 3.51 ± 0.76 | 18.99 ± 6.58 | 73.65 ± 27.71 | 39.23 ± 10.11 | 134.75 ± 44.25 |
| $M_{PB}/M_P$, % | 45.8 ± 3.2 | 74.9 ± 4.2 | 100 | 100 | 92.64 ± 2.0 |
| | Transition zone | | | | |
| $N_{PB}$ | 3.1 ± 0.6 | 2.1 ± 0.6 | 2.1 ± 0.4 | 0 | 7.3 ± 1.0 |
| $N_{PB}/N_P$, % | 68.8 ± 13.7 | 80.8 ± 10.3 | 100 | 0 | 79.3 ± 7.8 |
| $M_{PB}$ | 0.345 ± 0.107 | 1.79 ± 0.476 | 15.62 ± 7.94 | 0 | 17.75 ± 8.21 |
| $M_{PB}/M_P$, % | 61.7 ± 10.5 | 83.2 ± 13.6 | 100 | 0 | 96.75 ± 12.9 |
| | Western shelf | | | | |
| $N_{PB}$ | 1.0 ± 0.5 | 1.4 ± 0.8 | 0.2 ± 0.1 | 0 | 2.6 ± 1.4 |
| $N_{PB}/N_P$, % | 66.7 ± 14.4 | 93.4 ± 13.4 | 100 | 0 | 81.2 ± 11.2 |
| $M_{PB}$ | 0.14 ± 0.03 | 1.49 ± 0.39 | 0.19 ± 0.11 | 0 | 1.82 ± 0.42 |
| $M_{PB}/M_P$, % | 61.6 ± 12.3 | 90.96 ± 6.4 | 100 | 0 | 88.5 ± 9.9 |
| | Continental slope | | | | |
| $N_{PB}$ | 0.07 ± 0.02 | 0.02 ± 0.01 | 0 | 0 | 0.09 ± 0.02 |
| $N_{PB}/N_P$, % | 71.6 ± 8.7 | 60.0 ± 11/0 | 0 | 0 | 68.1 ± 9.2 |
| $M_{PB}$ | 0.011 ± 0.005 | 0.025 ± 0.012 | 0 | 0 | 0.036 ± 0.012 |
| $M_{PB}/M_P$, % | 78.4 ± 16.2 | 58.2 ± 12.0 | 0 | 0 | 76.2 ± 10.2 |

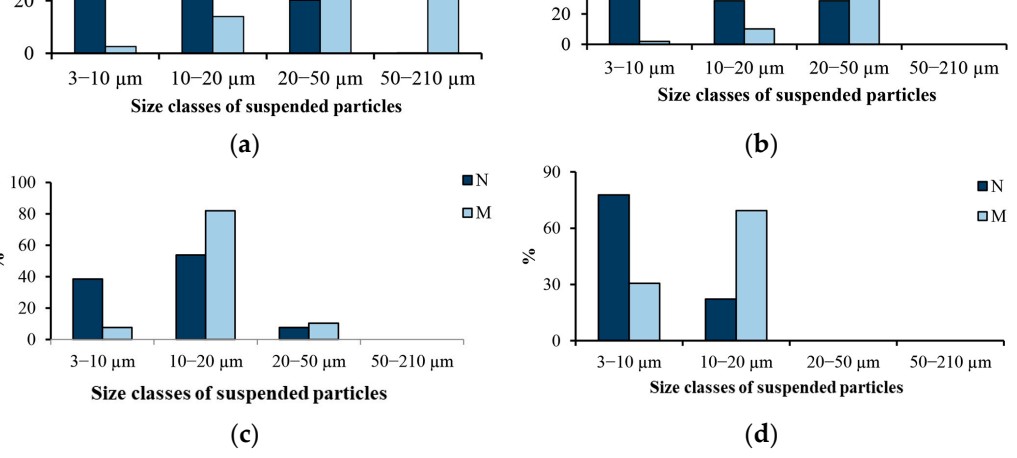

**Figure 3.** Proportion (%) of different size classes of suspended particles with attached bacteria in the total number (N) and mass (M) of suspended particles with attached bacteria in the water column of (**a**) Khatanga Bay, (**b**) transition zone, (**c**) Western shelf, and (**d**) continental slope of the Western Laptev Sea.

*3.2. Bacterial Abundance and Biomass*

The total abundance ($N_B$) and total biomass ($B_B$) of bacterioplankton on the transect varied, respectively, from $80.8 \times 10^3$ cell $mL^{-1}$ up to $2311.8 \times 10^3$ cell $mL^{-1}$ and from 1.3 to 37.7 mg C $m^{-3}$ (Figure 2). The average $N_B$ and $B_B$ decreased in the direction from the Khatanga River estuary to the open sea (Table 4). The average volume of a bacterial cell on the transect varied within 0.044–0.103 $μm^3$, averaging for the water column at stations: in

KHAB, $0.078-0.082$ $\mu m^3$; in the TZ, $0.086$–$0.087$ $\mu m^3$; on the WS, $0.058$–$0.068$ $\mu m^3$; and on the CS, $0.065$ $\mu m^3$.

**Table 4.** Mean $\pm$SE (min–max) total abundance ($N_B$, $10^5$ cells mL$^{-1}$), biomass ($B_B$), and contribution (%) of different groups of bacteria in the formation of $N_B$ and $B_B$.

| Stations | Parameters | | Proportion (%) of Different Groups of Bacteria | | |
| --- | --- | --- | --- | --- | --- |
| | | | Free-Living | Particle-Attached | In Microcolony |
| 5627 | $N_B$ | 16.2 ± 1.2 (13.4–18.8) | 31.3 ± 7.(23.3–47.8) | 63.0 ± 7.6 (47.7–79.8) | 5.7 ± 3.1 (0–13.1) |
| | $B_B$, mg m$^{-3}$ | 130.3 ± 15.3 (105.4–167.2) | 31.4 ± 6.7 (23.0–47.8) | 53.6 ± 9.5 (37.0–76.7) | 15.0 ± 10.2 (0–30.6) |
| | $B_B$, mg C m$^{-3}$ | 30.8 ± 2.2 (27.9–36.3) | 31.7 ± 6.7 (22.3–48.0) | 56.5 ± 8.9 (44.6–77.7) | 11.8 ± 7.7 (0–30.6) |
| 5630 | $N_B$ | 18.8 ± 1.3 (16.6–23.1) | 50.1 ± 3.0 (39.6–55.1) | 48.6 ± 3.4 (43.8–60.4) | 1.3 ± 0.9 (0–4.5) |
| | $B_B$, mg m$^{-3}$ | 139.4 ± 15.3 (108.6–190.1) | 62.2 ± 3.8 (50.2–71.4) | 36.0 ± 4.9 (22.1–49.8) | 1.8 ± 1.4 (0–6.5) |
| | $B_B$, mg C m$^{-3}$ | 32.4 ± 1.7 (28.1–37.7) | 56.9 ± 3.6 (47.2–66.6) | 41.3 ± 4.6 (27.4–52.8) | 1.8 ± 1.2 (0–6.0) |
| 5631 | $N_B$ | 10.2 ± 1.5 (8.0–15.4) | 64.2 ± 3.3 (57.7–75.2) | 35.8 ± 3.4 (24.8–42.3) | 0 |
| | $B_B$, mg m$^{-3}$ | 87.4 ± 11.2 (73.5–126.0) | 66.8± 5.2 (52.1–81.6) | 33.2 ± 4.8 (18.4–47.9) | 0 |
| | $B_B$, mg C m$^{-3}$ | 20.8 ± 2.7 (16.7–30.4) | 66.0 ± 5.1 (53.4–79.9) | 34.0 ± 4.7 (20.1–46.6) | 0 |
| 5632 | $N_B$ | 5.8 ± 0.6 (4.6–7.5) | 66.8 ± 1.7 (62.1–83.4) | 31.2 ± 2.8 (8.7–37.9) | 2.0 ± 1.3 (0–7.9) |
| | $B_B$, mg m$^{-3}$ | 44.4 ± 4.5 (38.9–60.0) | 73.9 ± 2.4 (66.5–84.3) | 22.5 ± 3.2 (6.1–33.5) | 3.6 ± 2.6 (0–13.2) |
| | $B_B$, mg C m$^{-3}$ | 10.9 ± 1.0 (9.3–14.5) | 72.7 ± 1.5 (65.5–82.4) | 24.2 ± 2.9 (6.8–34.5) | 3.1 ± 2.2 (0–11.6) |
| 5591-2 | $N_B$ | 2.4 ± 0.8 (1.0–4.3) | 84.5 ± 8.3 (61.0–98.2) | 15.5 ± 8.3 (1.8–39.0) | 0 |
| | $B_B$, mg m$^{-3}$ | 17.9 ± 9.1 (5.9–40.1) | 86.8 ± 6.2 (69.5–98.8) | 13.2± 6.2 (1.2–30.5) | 0 |
| | $B_B$, mg C m$^{-3}$ | 4.3 ± 2.1 (1.3–9.5) | 88.2 ± 4.1 (76.9–96.8) | 11.8 ± 4.1 (3.2–23.1) | 0 |
| 5590-2 | $N_B$ | 2.4 ± 0.8 (1.4–4.4) | 94.1 ± 1.9 (90.8–98.6) | 5.9 ± 1.9 (1.4–9.2) | 0 |
| | $B_B$, mg m$^{-3}$ | 12.2 ± 2.4 (8.3–18.0) | 93.7 ± 2.0 (90.6–99.0) | 6.3± 2.1 (1.0–9.4) | 0 |
| | $B_B$, mg C m$^{-3}$ | 3.1 ± 0.6 (2.2–4.5) | 92.9 ± 2.0 (88.9–98.8) | 7.1 ± 2.4 (1.2–11.4) | 0 |
| 5634 | $N_B$ | 2.5 ± 0.4 (0.9–4.0) | 95.1 ± 2.5 (83.2–100) | 3.8 ± 2.4 (0–16.8) | 1.1 ± 0.9 (0–6.7) |
| | $B_B$, mg m$^{-3}$ | 16.7 ± 2.8 (6.3–28.0) | 94.5 ± 2.8 (81.2–100) | 4.2 ± 2.7 (0–18.8) | 1.3 ± 1.1 (0–7.8) |
| | $B_B$, mg C m$^{-3}$ | 4.25 ± 0.7 (1.6–7.0) | 94.7 ± 2.8 (81.5–100) | 4.1 ± 2.6 (0–18.5) | 1.2 ± 1.0 (0–7.1) |
| 5635 | $N_B$ | 2.1 ± 0.3 (0.8–2.8) | 97.7 ± 0.7 (95.2–100) | 1.0 ± 0.4 (0–3.0) | 1.3 ± 0.5 (0–3.6) |
| | $B_B$, mg m$^{-3}$ | 13.4 ± 1.6 (6.8–18.3) | 96.8± 1.1 (92.7–100) | 1.1 ± 0.5 (0–3.5) | 2.1 ± 0.9 (0–7.1) |
| | $B_B$, mg C m$^{-3}$ | 3.6 ± 0.4 (1.6–4.7) | 97.1 ± 0.9 (93.9–100) | 1.1 ± 0.5 (0–3.4) | 1.8 ± 0.8 (0–5.8) |

Bacterioplankton included free-living bacteria, particle-attached bacteria, and bacteria in microcolonies (Table 3). In the inner part of Khatanga Bay (station 5627), the abundance and biomass of particle-attached bacteria prevailed over other groups (Table 4). The share of particle-attached bacteria in NB correlated positively with the amount of DYP ($r = 0.92$, $p < 0.001$) and negatively with S ($r = -0.835$, $p < 0.001$). The average abundances of bacteria on the visible surface of colonized DYP of different sizes were as follows: 3–10 $\mu m$, 4.0 ± 0.6 (range 2–10) cells per particle; >10–20 $\mu m$, 7.5 ± 1.0 (range 3–24) cells per particle; >20–50 $\mu m$, 11.5 ± 1.8 (range 4–28) cells per particle; >50–120 $\mu m$, 657 ± 69 (range 114–1280) cells per particle; and >120–210 $\mu m$, 1604 ± 202 (range $1333-2300$) cells per particle.

Moving from the inner part of Khatanga Bay to the open sea at $N_B$ and $B_B$, the share of free-living bacteria increased, which in the area of the CS (station 5635) was about 97% (Table 4).

The maximum number of microcolonies (4486 colonies mL$^{-1}$) was found at station 5627 at a depth of 5 m; the minimum (523 colonies mL$^{-1}$) was found at station 5635 at depths of 0 m and 850 m. The abundance and biomass of bacteria in microcolonies were highest in Khatanga Bay (Table 4), where the number of cells in a microcolony varied from 14 to 128 cells per colony. The sizes of bacterial cells in microcolonies ranged from 0.8 × 0.5 to 1 $\mu m$. The number of cells in microcolonies in the TZ ranged from 14 to 40 cells per colony, while on the WS and CS, it did not exceed 19 cells per colony. Filamentous bacteria from 6.0 × 0.4 to 8.0 × 0.4 $\mu m$ in size were observed at stations 5627 and 5631, but they constituted only a small fraction (<0.1%) of the total abundance and biomass.

Bacterial abundance and biomass were positively correlated with temperature, DOC, $N_P$, $M_P$, and $NH_4^+$ in the water column (Table 5). There was a significant inverse relationship between salinity, alkalinity, and concentration of bacterioplankton, and weak negative correlations between the concentration of bacterioplankton and $PO_4^{3-}$ (Table 5).

**Table 5.** Spearman correlation coefficients ($r_S$) with corresponding *p* values and number of paired data (*n*) between bacterial and environmental parameters.

| Parameters (Units) | $N_B$ ($10^3$ Cells mL$^{-1}$) | $B_B$ (mg C m$^{-3}$) | $P_B$ ($10^3$ Cells mL$^{-1}$ d$^{-1}$) | $P_B$ (mg C m$^{-3}$ d$^{-1}$) |
|---|---|---|---|---|
| T (°C) | 0.828 * | 0.800 * | 0.834 * | 0.823 * |
| S (psu) | −0.814 * | −0.781 * | −0.848 * | −0.833 * |
| Alk (mg eq L$^{-1}$) | −0.823 ** | −0.790 ** | −0.819 ** | −0.795 ** |
| DOC (μM) | 0.799 * | 0.782 * | 0.813 * | 0.817 * |
| $NH_4^+$ (μM) | 0.743 * | 0.739 * | 0.758 * | 0.757 * |
| $PO_4^{3-}$ (μM) | −0.386 *** | −0.372 *** | −0.432 *** | −0.424 *** |
| $N_P$ ($10^3$ particles mL$^{-1}$) | 0.665 * | 0.652 * | 0.767 * | 0.726 * |
| $M_P$ (mg m$^{-3}$) | 0.579 * | 0.637 * | 0.696 * | 0.734 * |
| *n* | 34 | 34 | 25 | 25 |

$N_B$—Bacterial abundance, $B_B$—Bacterial biomass, $P_B$—Bacterial production, T—temperature; S—salinity; DOC—dissolved organic carbon; Alk—Alkalinity; $N_P$—abundance of suspended particles; $M_P$—mass of suspended particles; $NH_4^+$—ammonia; $PO_4^{3-}$—phosphate. Levels of significance: * $p < 0.001$, ** $p < 0.01$ and *** $p < 0.05$.

### 3.3. Bacterial Production

The bacterial growth rates (μ) and doubling period on the transect varied within $0.1726-0.3574$ d$^{-1}$ and 1.9–4.2 d, averaging for the water column in KHAB $0.3355 \pm 0.007$ d$^{-1}$ and $2.07 \pm 0.04$ d; in the TZ, $0.2877 \pm 0.0165$ d$^{-1}$ and $2.46 \pm 0.15$ d; on the WS, $0.2066 \pm 0.0107$ d$^{-1}$ and $3.26 \pm 0.19$ d; and on the CS, $0.2129 \pm 0.0069$ d$^{-1}$ and $3.27 \pm 0.10$ d (Table 6). Bacterioplankton production varied widely (Table 6). The minimum and maximum production values ($P_B$) differed by 42 times (in cell mL$^{-1}$ d$^{-1}$) and 64 times (in mg C m$^{-3}$ d$^{-1}$). The average $P_B$ values for the water column decreased in the direction from the river estuary to the open sea: KHAB, $(561.5 \pm 26.5) \times 10^3$ cell mL$^{-1}$ d$^{-1}$ and $10.3 \pm 0.6$ mg C m$^{-3}$ d$^{-1}$; in the TZ, $(282.2 \pm 32.5) \times 10^3$ cell mL$^{-1}$ d$^{-1}$ and $5.65 \pm 0.6$ mg C m$^{-3}$ d$^{-1}$; on the WS, $(52.7 \pm 11.7) \times 10^3$ cell mL$^{-1}$ d$^{-1}$ and $0.85 \pm 0.24$ mg C m$^{-3}$ d$^{-1}$; on the CS, $(42.7 \pm 8.8) \times 10^3$ cell mL$^{-1}$ d$^{-1}$ and $0.7 \pm 0.1$ mg C m$^{-3}$ d$^{-1}$. Strong positive correlations were noted between $P_B$ and T °C and between $P_B$ and DOC (Table 5). Between $P_B$ and S, strong negative correlations were observed (Table 5).

**Table 6.** Bacterial growth rates (μ), doubling period (D), and production ($P_B$) of bacterioplankton.

| Station | Depth | μ, Day$^{-1}$ | D, Day | $P_B$, $10^3$ Cells mL$^{-1}$ d$^{-1}$ | $P_B$, mg C m$^{-3}$ d$^{-1}$ |
|---|---|---|---|---|---|
| | | | Khatanga Bay | | |
| | 0 | 0.3406 | 2.04 | 457.7 | 9.5 |
| 5627 | 5 | 0.3574 | 1.94 | 580.8 | 12.9 |
| | 14 | 0.3348 | 2.07 | 628.3 | 9.4 |
| | 0 | 0.3522 | 1.97 | 647.7 | 11.6 |
| 5630 | 20 | 0.3088 | 2.24 | 525.0 | 8.7 |
| | 23 | 0.3195 | 2.17 | 529.4 | 9.8 |
| | | | Transition zone | | |
| | 0 | 0.3400 | 2.04 | 307.4 | 6.4 |
| 5631 | 18 | 0.2843 | 2.44 | 436.6 | 8.6 |
| | 25 | 0.2504 | 2.77 | 201.3 | 4.3 |
| | 0 | 0.3252 | 2.13 | 242.7 | 4.7 |
| 5632 | 17 | 0.3026 | 2.29 | 296.8 | 5.7 |
| | 22 | 0.2238 | 3.10 | 208.1 | 4.2 |

**Table 6.** *Cont.*

| Station | Depth | $\mu$, Day$^{-1}$ | D, Day | $P_B$, $10^3$ Cells mL$^{-1}$ d$^{-1}$ | $P_B$, mg C m$^{-3}$ d$^{-1}$ |
|---|---|---|---|---|---|
| | | | Western shelf | | |
| | 0 | 0.2751 | 2.52 | 118.0 | 2.6 |
| 5591-2 | 13 | 0.1879 | 3.69 | 32.7 | 0.4 |
| | 41 | 0.1794 | 3.86 | 18.5 | 0.2 |
| | 0 | 0.2115 | 2.44 | 92.9 | 0.9 |
| 5590-2 | 13 | 0.1726 | 2.68 | 24.8 | 0.4 |
| | 59 | 0.2245 | 3.81 | 31.4 | 0.6 |
| | 0 | 0.2115 | 3.28 | 85.7 | 1.5 |
| 5634 | 40 | 0.1726 | 4.02 | 15.1 | 0.3 |
| | 182 | 0.2245 | 3.09 | 55.2 | 0.8 |
| | | | Continental slope | | |
| | 0 | 0.2229 | 3.11 | 42.1 | 0.6 |
| | 27 | 0.1950 | 3.55 | 15.7 | 0.3 |
| 5635 | 140 | 0.2044 | 3.39 | 48.5 | 0.8 |
| | 850 | 0.2294 | 3.02 | 64.6 | 1.1 |

## 4. Discussion

The Western Laptev Sea is impacted by Khatanga River freshwater runoff. The river water reduces the surface salinity in Khatanga Bay to 3.5–5.0 psu and increases the temperature to 3.9–4.2 °C. In September 2017, the primary production values for the water column along the transect from Khatanga Bay to the continental slope were low (Table 7), which is explained by the low level of incident radiation and low nutrient concentrations [51,54]. At the same time, relatively high concentrations of DOC (162–728 µM) were observed in the Western Laptev Sea during this period (Table 1). The higher DOC concentration measured for the upper water layer in the mixing zone formed under the influence of Katanga River runoff [36]. Breakdown of dead algae cells, production of marine biota, and reworking of DOM were considered possible autochthonous sources of fluorescent dissolved organic matter in the Siberian shelf seas in September 2017 [55].

**Table 7.** Biological properties at stations in the Western Laptev Sea.

| Station | Photic Zone *, m | Chl$_0$ [a] | Chl$_{FL}$ [b] | B$_{PH}$ [c] | B$_{PPH}$ [d] | PP$_0$ [a] | PP$_{FL}$ [a] | B$_{HF}$ [e] | B$_Z$ [f] |
|---|---|---|---|---|---|---|---|---|---|
| 5627 | 7 | 1.65 | 1.9 | 15.7 | 3.42 | 8.50 | 17 | 7.7 | 20.8 |
| 5630 | 8 | 0.82 | 0.9 | 23.0 | 0.73 | 11.96 | 21 | 3.6 | 4.5 |
| 5631 | 8 | 1.42 | 0.9 | 12.7 | 0.58 | 22.88 | 49 | 4.2 | 5.9 |
| 5632 | 15 | 0.44 | 0.2 | 3.7 | 1.15 | 6.25 | 28 | 4.2 | 6.6 |
| 5591-2 | 32 | 0.27 | 0.1 | 2.4_ | 1.34 | 3.43 | 32 | 1.4 | 11.2 |
| 5590-2 | 35 | 0.24 | 0.2 | 0.9 | 1.61 | 1.66 | 20 | 1.0 | 20.9 |
| 5634 | 32 | 0.25 | 0.2 | 1.3 | 1.94 | 1.62 | 16 | 0.9 | 21.5 |
| 5635 | 35 | 0.27 | 0.4 | 14.6 | 0.74 | 2.38 | 17 | 1.0 | 0.9 |

*—Based on measured 1% light penetration depth (Demidov et al., 2019 [51]); [a]—Data from Demidov et al., 2019 [51]; [b]—Data from Demidov et al., 2020 [54]; [c]—Data from Sukhanova et al., 2019 [35]; [d]—Data from Belevich et al., 2021 [56]; [e]—our dates; [f]—Data from Pasternak et al., 2022 [57]; Chl$_0$—chlorophyll *a* concentration (mg m$^{-3}$) in the surface layer; Chl$_{FL}$—mean concentration of chlorophyll *a* (mg m$^{-3}$) in the photic zone; B$_{PH}$—mean biomass of phytoplankton (mg C m$^{-3}$) in the photic layer; B$_{PPH}$—mean biomass of picophytoplankton (mg C m$^{-3}$) in the photic layer; PP$_0$—primary production (mg C m$^{-3}$ d$^{-1}$) in the surface layer; PP$_{FL}$—integral primary production (mg C m$^{-2}$ d$^{-1}$) in the photic layer; B$_{HF}$—mean biomass of heterotrophic flagellates (mg C m$^{-3}$) in water column (our dates); B$_Z$—mean biomass of zooplankton (mg C m$^{-3}$) in water column.

During the study period, a positive correlation ($r = 0.75$, $p < 0.05$, $n = 8$) between the chlorophyll a concentration and bacterioplankton biomass in the surface water layer was observed, but no correlation between the integral values of these parameters for the photic zone. Only a weak positive correlation was revealed between the integral values of the phytoplankton (B$_{PH}$ + B$_{PPH}$, Table 7) and bacterioplankton biomass for the photic

zone. In the surface water layer, a strong positive correlation was established between the primary production of phytoplankton and bacterioplankton (r = 0.83, *p* < 0.05, *n* = 8), but no relationship was found between the integral values of these parameters for the photic zone.

Heterotrophic nanoflagellates (HNF) were observed in all samples (Table 7). The abundance of HNF varied from 70 cells mL$^{-1}$ (continental slope) to 636 cells mL$^{-1}$ (Khatanga Bay), averaging 205 ± 25 cells mL$^{-1}$. The ratio of bacteria abundance to HNF abundance (1225−9469, 3089 ± 319 on average) indicated favorable trophic conditions for the development of bacterivorous HNF. The HNF biomass was 8.1–53.8 (on average 27.4 ± 1.9) % of the bacterioplankton biomass. Strong positive correlations were found between the abundance of bacteria and the abundance of HNF, as well as between the biomass of bacteria and the biomass of HNF: respectively, r = 0.73, *p* < 0.001, *n* = 34, and r = 0.82, *p* < 0.001, *n* = 34. Apparently, at the end of the growing season, HNFs play an essential role as regulators of bacterial abundance in the water column.

### 4.1. DAPI-Positive Yellow Suspended Particles

Together with runoff, a large amount of DAPI-positive yellow suspended particles (DYP) enters the sea. On average, the mass of organic suspended particles with attached microorganisms and viruses in Khatanga Bay, the TZ, and on the WS exceeded the plankton biomass in raw matter, respectively, by 300, 89, and 6 times, whereas in the CS area, it was 1.5 times lower. Note that in KHAB, the TZ, and estuaries of other Siberian rivers, viruses and HNF are also attached to suspended particles [58,59]. These particle-based microbial communities may likely be a food source for higher trophic levels [19].

In surface waters from the Humber estuary, the concentration of SPM in September was 138–274 g m$^{-3}$. As well, 58–77% of particles were in the 5–14 μm size range; 12–30%, <5 μm; and only 2–13% were >24 μm [60]. In water samples taken at a station 500 m off the coast of Bilbao, Spain, the number of particles >3 μm reached 39.5 × 10$^3$ particles mL$^{-1}$ [15]. The share of particles with bacterial cells was 52.0–68.1%. "In particles on which bacteria were observed, the average number of bacteria fluctuated between 2.9 and 6.4 bacteria per particle" [15]. "Bacteria are a high-quality food source, and when they are attached to particles, they increase the quality of detrital material as food" [61]. "Particle-attached bacteria can play a very different role in a food web than free-living bacteria, because they may be directly grazed upon by larger metazoans, bypassing consumption by protozoan grazers and short-circuiting the microbial loop" [62]. "Detritivores metazoans may therefore be the principle consumers of bacterial biomass in estuaries" [63].

### 4.2. Bacterial Abundance and Biomass

Water masses along our transect from Khatanga Bay to the Western shelf and continental slope of the Laptev Sea differed greatly in composition, abundance, biomass, and production of the bacterial community, which in turn were largely controlled by the availability, temperature, DOC, and abundance of suspended particles. We found large spatial variations in the contribution of particle-attached bacteria to the total bacterial abundance and biomass. The maximum values (79.8% N$_B$ and 77.7% B$_B$) were found in the inner part of Khatanga Bay, and the minimum (0.1% N$_B$ and 0.1% B$_B$) in the continental slope area at depths of 140–320 m. Particle-attached bacteria are an important structural component of the planktonic community of Khatanga Bay.

According to the literature data, in coastal waters, the abundance of bacteria attached to suspended particles can vary from a few percent to 98% of the total bacterial abundance in various aquatic regions, mainly depending on particle abundance [15,62].

As in other Arctic regions [2,25,64], the total bacterial abundance and biomass in the Western Laptev Sea tend to decrease from the coast to the central Arctic Ocean. In our study, the maximum N$_B$ and B$_B$ values were close to those found in Arctic coastal waters (1.1–4.1) × 10$^6$ cells mL$^{-1}$ [65–70]; the minimum is typical of deeper regions of the Arctic Ocean (0.7–9.1) × 10$^6$ cells mL$^{-1}$ [25,64,70–73]. The average volume of a bacterial cell in the Laptev Sea is close to that in other areas of the Arctic (Chukchi Sea and Canada Basin),

where it is approximately $0.05-0.09$ μm$^3$ [74]. In the Arctic Ocean, the bacterioplankton biomass varies from 5 mg C m$^{-3}$ in the Central Ocean [75] to 100 mg C m$^{-3}$ in Disko Bay, Western Greenland [65]. The bacterial biomass from the Western Laptev Sea is within the range of bacterial biomass reported from Arctic marine ecosystems [65,73,75].

The phytoplankton and bacterial biomass in the photic zone of Arctic ecosystems were compared mainly in the spring and summer.

The integrated bacterial biomass in the Northern Barents Sea in the photic zone during the development of the spring bloom corresponded to 1–26% of the phytoplankton biomass [73]. On the Western coast of Greenland during and after a diatom bloom (June), the bacterial and phytoplankton biomass ratio varied from 45–95% [65]. At the end of summer, in this region, the integrated bacterioplankton biomass was 26–58% of the integrated phytoplankton biomass [76]. The bacterial and phytoplankton biomass ratios obtained in our study in autumn differ from those found in spring and summer in areas unaffected by river runoff (Figure 4). In Khatanga Bay and the Southern sector of the transition zone, the integral bacterioplankton biomass in the photic zone exceeded that of phytoplankton by a factor of 1.4; in the Northern sector of the transition zone and the Western shelf, the difference increased to 1.5–2.0 times. However, in the continental slope area, the integral bacterioplankton biomass was only 18.1% of the integral phytoplankton biomass (Figure 4). These ratios largely depended on the distribution of phytoplankton biomass on the transect. In autumn, the average phytoplankton biomass in the zone of influence of Khatanga River runoff was relatively high (89–160 mg m$^{-3}$); on the shelf, they decreased significantly (up to 8–30 mg m$^{-3}$), but in the area of the continental slope, they again increased significantly (up to 150 mg m$^{-3}$) [35]. The continental slope area in the Laptev Sea is a specific local biotope where a phytocenosis has developed with the formation of deep maxima, differing from adjacent areas. Deep phytoplankton maxima can occur due to the high water transparency, the large vertical extent of the euphotic layer, the absence of a rigid pycnocline, and the availability of nutrients [35]. In September 1991, in the ecosystem of the Lena River estuary (Eastern Laptev Sea), during the period of mass development of the symbiotic ciliates *Mesodinium rubrum* (Lohmann, 1908), the phytoplankton biomass was 12 times less than that of bacterioplankton, but taking into account the biomass of *M. rubrum*, the biomasses of planktonic eukaryotic autotrophic and prokaryotic heterotrophic microorganisms were comparable [32].

Analysis of the results of our studies and data on assessing the biomass of phytoplankton [35] and zooplankton [57] obtained for the Western Laptev Sea in September 2017 showed that the total biomass of the plankton community per unit volume of water on the KHAB–CS transect decreased by an order of magnitude (Figure 5). At the end of the growing season in the Western Laptev Sea, with the exception of the shelf, bacterioplankton was the main component (44.5–55.2%) of the plankton community biomass. The high biomass of the microbial community (bacteria and HNF) and observed bacterial activity suggested that planktonic microbial communities play a major role in pelagic carbon turnover at the end of the growing season.

In the Eastern Laptev Sea, which receives Lena River runoff, similar data were obtained in September 2015: the share of bacterioplankton in the total biomass of the plankton community on the inner, middle, and outer shelf and in the epipelagic zone of the deep-water area 54.5, 49.2, 33.5, and 30.0%, respectively [58]. In Disko Bay, Western Greenland, in June–July 1992, during the diatom bloom, the total biomass of the plankton community varied between 190 and 252 mg C/m$^3$, and the share of bacterioplankton in the total biomass was 24% [65].

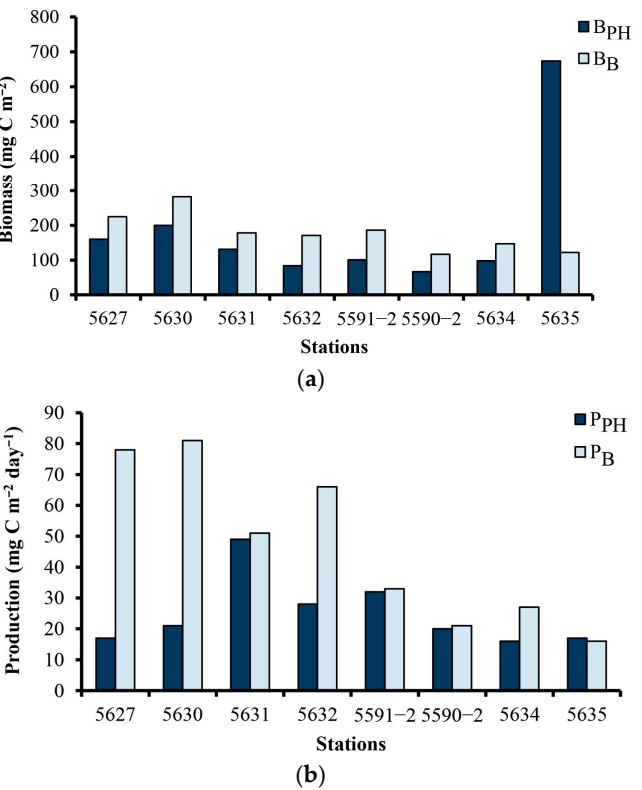

**Figure 4.** Comparison of biomass and production of phytoplankton and bacterioplankton. (**a**) phytoplankton ($B_{PH}$) and bacterioplankton ($B_B$) biomasses (mg C m$^{-2}$) integrated in the photic zone; (**b**) phytoplankton ($P_{PH}$) and bacterioplankton ($P_B$) production (mg C m$^{-2}$ day$^{-1}$), integrated in the photic zone.

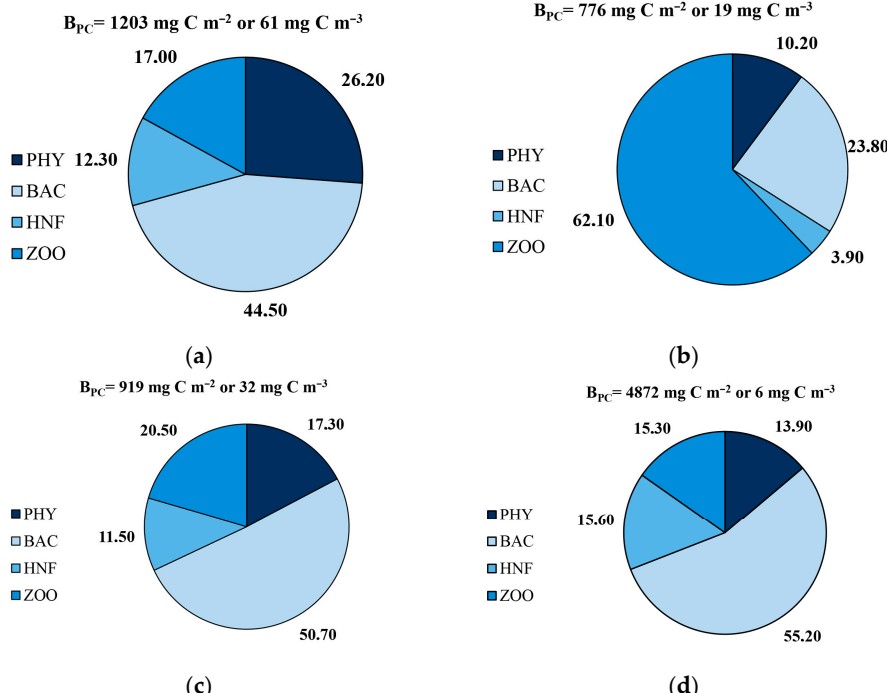

**Figure 5.** Proportion (%) of phytoplankton (PHY), bacterioplankton (BAC), heterotrophic nanoflagellates (HNF), and zooplankton (ZOO) in the total plankton community biomass ($B_{PC}$, mgC m$^{-2}$ or mgC m$^{-3}$) in (**a**) Khatanga Bay, (**b**) transition zone, (**c**) Western shelf, and (**d**) continental slope of the Western Laptev Sea.

### 4.3. Bacterial Production

One of our findings is that mean bacterial production in the water column in Khatanga Bay ($10.3 \pm 0.6$ mg C m$^{-3}$ d$^{-1}$) was about two times higher than in the TZ and about 12 times higher than on the shelf and near the CS, indicating a direct relationship between bacterial activity and DOC (and temperature), as well as an inverse relationship between bacterial activity and salinity. Earlier, Saliot et al. [31] and Sorokin and Sorokin [32] reported higher bacterial production in the Lena River Delta and decreasing production in the adjacent Laptev Sea. The bacterioplankton production documented in the present study is within the range reported for coastal areas and estuaries, 0.3–115.9 mg C m$^{-3}$ d$^{-1}$ [19,27,69,77], and for open areas, 0.07–41.1 mg C m$^{-3}$ d$^{-1}$ [74,77] of the Arctic Ocean.

In our study, growth rates were highly variable, although within the range (<0.05–0.6 day$^{-1}$) of previously published data [28,67,72,78,79]. High rates of bacterial production at the end of the growing season have been found in the Chukchi and Greenland seas [67,79]. In the upper 50 m of the water column in the Greenland Sea, bacterial counts averaged $1.1 \times 10^6$ cells mL$^{-1}$, bacterial growth rate averaged 0.68 d$^{-1}$, and bacterial production averaged 0.26 µM C d$^{-1}$ [67]. High growth rates were also observed in waters with temperatures below 0 °C; for example, a growth rate of 0.31 corresponds to a generation time of 2 d at $-1.3$ °C at 50 m. "The high bacterial growth and production rates may be related to fact that at the end of the productive season, considerable amounts of annually produced transient organic matter were present" [67].

During cruise 69 of the R/V Akademic Mstislav Keldysh, Demidov et al. [51] measured phytoplankton primary production in parallel with our bacterial production measurements. Thus, we can combine our data to estimate the metabolic balance of the system. In September in the Western Laptev Sea in Khatanga Bay, the integral bacterial production exceeded the integral primary production in the photic zone by 4.3–4.5 times, and the ratio of bacterial production to primary production in other parts of the Western Laptev Sea most often fluctuated around 1 (Figure 4). During the development of the spring bloom, the bacterial production to primary production ratios in the photic zone varied from 3 to 97% in the Franz Joseph Land archipelago [80] and from 8 to 143% in the Northern Barents Sea [73]. In August in the Central Arctic Ocean, the ratio of bacterial production (based on $^{14}$C-leucine incorporation) to particulate phytoplankton production increased from 0.1 in open water to 2.43 at ice-covered stations [79]. In September 1991, in the Lena Estuary ecosystem (Eastern Laptev Sea), bacterioplankton production was twice as high as primary phytoplankton production [32].

### 4.4. Role of Bacteria in the Carbon Budget of the Western Laptev Sea

Our results allow a first-order estimate of bacterial carbon fluxes for the Western Laptev Sea in autumn. The values of bacterioplankton production obtained in this study make it possible to estimate bacterial carbon demand (BCD) and bacterial respiration (BR) in different parts of the Western Laptev Sea. Bacterial growth efficiency (BGE), the ratio of biomass produced to substrate assimilated, can be used to estimate the total bacterial carbon demand. BGE is computed as [81] BGE = BP/(BP + BR), where BP is bacterial production and BR is bacterial respiration. Rearrangement of the equation yields an estimate of bacterial respiration: BR = (BP/BGE) − BP. In our study, we used a BGE of 0.27 [49]. The integral BCD values in the photic zones of the studied areas of the sea varied from 299 mg C m$^{-2}$ day$^{-1}$ in Khatanga Bay to 50−58 mg C m$^{-2}$ day$^{-1}$ on the outer shelf and continental slope. As a result, they exceeded the integral primary production in Khatanga Bay by an order of magnitude; in other areas, by three to five times. Similar results were received in the Ob and Yenisei rivers, estuaries of rivers, and the Kara Sea, where mean rates of areal primary production in the euphotic zone are about two to four times lower than BCD [49]. Apparently, in the Laptev Sea, just like in the Kara Sea system, in autumn, primary production may be completely consumed by heterotrophic processes with little surplus production remaining for export from the shelf to the Central Arctic Ocean [49]. According to our estimates, during the study period, the daily consumption of

DOC by bacterioplankton, or BCD, was $0.63 \pm 0.06\%$ in Khatanga Bay, $0.47 \pm 0.05\%$ in the TZ, and $0.09 \pm 0.02\%$ of the DOC concentration on the WS. Consequently, in these areas, $0.46 \pm 0.05\%$, $0.34 \pm 0.04\%$, and $0.07 \pm 0.01\%$, respectively, of the DOC content in water per day should be broken down (mineralized) by bacterioplankton for respiration.

Many estuaries are net heterotrophic ecosystems that emit $CO_2$ into the atmosphere [82–84]. The prevalence of heterotrophic metabolism and $CO_2$ efflux from aquatic ecosystems necessarily requires allochthonous inputs of organic carbon to the aquatic systems [85]. In our study, the $CO_2$ production rate ($P_{CO2}$) by bacteria in the photic zone averaged 236 mg C m$^{-2}$ d$^{-1}$ in Khatanga Bay; 158 mg C m$^{-2}$ d$^{-1}$ in the TZ; 73 mg C m$^{-2}$ d$^{-1}$ on the WS; and 42 mg C m$^{-2}$ d$^{-1}$ on the CS. $P_{CO2}$ exceeded the integral values of phytoplankton primary production in the photic zone in these areas by 12, 4, 3, and 2 times, respectively. In Summer 2004, in the Mackenzie River estuary (station R4, salinity 1 psu), depth-integrated bacterial respiration was 22 times higher than depth-integrated net algal primary production; in the TZ (station R5d, salinity about 4 psu), by 1.2 times; and in the coastal Beaufort Sea (station R9, salinity >20 psu), by 1.3 times [86].

The bacterial $CO_2$ production rate in the surface water layer (0–1 m) was, on average, in Khatanga Bay at 28.6 mg C m$^{-2}$ d$^{-1}$, and on the Western shelf at 3.7 mg C m$^{-2}$ d$^{-1}$. The area of Khatanga Bay is approximately 8140 km$^2$ (length 220 km, average width 37 km). Let us assume that the area of the WS is equal to half the entire area of the CS of the Laptev Sea 460,000 km$^2$ [87], i.e., 230,000 km$^2$. As a result, bacterial $CO_2$ production calculated for the area of Khatanga Bay ($2.33 \times 10^3$ t C area$^{-1}$ day$^{-1}$) was equivalent to 27% of that calculated for the entire area of the WS ($8.51 \times 10^3$ t C area$^{-1}$ day$^{-1}$). Apparently, Khatanga Bay was a significant source of $CO_2$ in the atmosphere at the time of sampling.

"Microbial processes in polar systems are particularly sensitive to small fluctuations in their environment and have a potentially large impact on the carbon flux and other ecosystemic functions" [24]. The warming of the Arctic Ocean impacts POM and DOM in the Arctic region, which affect the local bacterial communities. Increased POM and DOM levels may increase the population, production, and metabolic activity of bacteria in Arctic Ocean seawater, thereby releasing a significant amount of $CO_2$ [70]. The results of studying bacterioplankton in the Western Laptev Sea may be useful in predicting further changes in bacterial communities should climate change lead to further increases in river discharge and related particle loads.

## 5. Conclusions

The results provide new information on the distribution of bacterial abundance and production in the poorly studied Khatanga region of the Laptev Sea at the end of the growing season. We have demonstrated that the influence of the Khatanga discharge on bacterioplankton composition was limited mainly to the Gulf and transition zone and was poorly recognized in the Western shelf and continental scope areas. The average bacterial total abundance and biomass in Khatanga Bay were almost an order of magnitude higher than those on the continental slope. The contribution of bacteria attached to suspended particles (3–210 μm in size) to the total bacterial biomass decreased from 56.5% in Khatanga Bay to 1.1% at the continental slope, and the relative importance of this particle-attached bacterial fraction was negatively correlated with salinity and positively correlated with the number of suspended particles. Except for the CS, in other areas, the estimated depth-integrated bacterial biomass in the photic zone was higher than the depth-integrated phytoplankton biomass in the photic zone. Apparently, bacterioplankton, including particle-attached bacteria, was an important food source for zooplankton in coastal areas and on the shelf of the Laptev Sea.

Heterotrophic bacterial activity was high across the transect from the interior of Khatanga Bay to the continental slope. Estimated depth-integrated bacterial respiration rates in Khatanga Bay were higher than the depth-integrated primary production rate by 12 times, while at marine stations, bacterial respiration rates were higher than depth-integrated primary production rates only by two to three times. Thus, at the end of the

growing season in the plankton communities of the Western Laptev Sea, heterotrophic processes prevailed over autotrophic ones.

**Supplementary Materials:** The following supporting information can be downloaded at: https://www.mdpi.com/article/10.3390/jmse11081573/s1, Table S1. Abundance of suspended particles of different size classes (NP, particles $mL^{-1}$) and mass of suspended particles of different size classes (MP, mg $m^{-3}$) along the transect; Table S2. Abundance ($10^3$ cels $mL^{-1}$) of free-living bacteria, particle-associated bacteria, bacteria in microcolony and total abundance of bacterioplankton; Table S3. Mean volume of bacterial cell ($\mu m^3$) of free-living bacteria, particle-associated bacteria, bacteria in microcolony; Table S4. Biomass [mg $m^{-3}$ (mg C $m^{-3}$)] of free-living bacteria, particle-associated bacteria, bacteria in microcolony and total biomass of bacterioplankton; Table S5. Abundance ($N_{HNF}$, cells $mL^{-1}$) and biomass ($B_{HNF}$, mg C $m^{-3}$).

**Author Contributions:** A.I.K., D.B.K. and A.F.S. contributed to the design of the study. D.B.K. and A.V.R. performed field data collection and field experiments. A.I.K., D.B.K., A.V.R. and E.A.Z. processed water samples and analyzed the data. A.I.K., A.F.S. and D.B.K. wrote the draft of the manuscript. All authors have read and agreed to the published version of the manuscript.

**Funding:** The work was carried out within the state assignment of IBIW RAS (topic no. 121051100102-2). The research was supported by the Russian Science Foundation, project no. 22-17-00011. Grant Disclosures: https://rscf.ru/project/22-17-00011/ (accessed on 16 July 2023).

**Data Availability Statement:** The original contributions presented in the study are included in the article/Supplementary Material. Further inquiries can be directed to the corresponding author.

**Acknowledgments:** Research workers from IBIW RAS are grateful to Mikhail V. Flint for the opportunity to work with the crew of the R/V Akademik Mstislav Keldysh, whose own contributions during field studies are greatly appreciated.

**Conflicts of Interest:** The authors declare that the research was conducted in the absence of any commercial or financial relationships that could be construed as a potential conflict of interest.

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
