# Peer review of "Abundance, Biomass, and Production of Bacterioplankton at the End of the Growing Season in the Western Laptev Sea: Impact of Khatanga River Discharge (Arctic)"

_jmse, doi:10.3390/jmse11081573_

Round 1

Reviewer 1 Report

This manuscript is a clean and clear report on the bacterial status in the Laptev Sea river plume. 

I recommend accept for publication after very minor revision.

Line 37, productivity of autotrophic and heterotrophic organisms. 

Figure 1 please show the Khatanga River, Khatanga Bay

Figure 2, c and d, color bar, delete the negative values because it is not reasonable. All the figures, maybe you could label some isolines to make it friendly to readers.

Figure 4, BPH in the figure and caption are in different format.  PPH also. 

Reviewer 2 Report

Review of “Abundance, Biomass, and Production of Bacterioplankton at the End of the Growing Season in the Western Laptev Sea: Impact of Khatanga River Discharge (Arctic)” by Kopylov Kosolapov, Romanenko , Zabotkina , and Sazhin

This manuscript reports on a series of measurements in a transect between riverine inputs and the Arctic Ocean taken in the Laptev Sea. The investigators measured bacterial abundance, particle association, microbial production and associated parameters to understand microbial significance in carbon (and nutrient) cycling in river input-rich Arctic environments. Bacteria were found to be significantly associated with particles near the river end-member and become much less so toward the ocean end-member. The authors used selected literature values for additional parameters (BGE primarily) to assess the overall bacterial contribution to carbon budget(s). Estimates for the entire shelf (summer) indicated that microbial DOM consumption and respiration leads to significant CO2 export.

The manuscript was reasonably well-written with a few noted inconsistencies from “traditional” English-language journals. The concepts and observations were clearly presented.

The manuscript offers significant information and a valuable contribution for a system that is understudied and arguably most vulnerable to climate change (potential increased inputs of DOM/POM and nutrients during longer summer seasons).

M&M: The section was overall well presented for many of the methods, but I did not see much (if any) information on how nutrient concentrations or DOC was measured? It’s hard to tell for sure, but some values were used from collaborative studies. I think this would be good to put in the M&M section; e.g. “DOC was measured as described previously [X, XX].”

Overall, I would suggest publication with minor edits.

Some specifics noted during my read:

Lns 32-34: Not sure why the first sentence is in quotes?

Ln 37: no “of”

Ln 42: “viral shunt” should be further explained. I believe what you mean is that the OM can be cycled from heterotrophic organisms back into the environment via viral lysis, but it’s no really clear to the reader by just adding, “viral shunt”.

Lns 45-46: this should be better qualified. The referenced work was not in the Arctic. Filter feeders in estuaries can be primary consumers (e.g. oysters, molluscs)

Lns 56-57: Awkward sentence. “have suggested” (past tense) and “is” are used in the same sentence. Take out “have”

Lns 59-61: not customary to directly quote prior manuscripts. Paraphrase in own words.

Table 1. Usually nitrite, nitrate, ammonium and phosphate are annotated with their charge (e.g. NO2-, NO3-, NH4+, PO43-).

Lns 131-132: Hard to understand. Sentence ends with “after”. After what? I think you mean counts made before and after sonication? Please clarify. After seeming Lns 144 and 145 – perhaps you mean (by “after”) in accordance with methods reported in the reference?

Lns 164-168: formatting is odd (centered?).

Figure 2: generally, the letter representing the graph is identified first, then described: e.g. a) temperature distribution ( C); (b) salinity (psu); (c) ….

See above comments. 

Reviewer 3 Report

Regarding the statistical analysis - we do not think that it is sufficient to use only the Spearman coefficients and we suggest that the authors use other tests, possibly to make comparisons between several statistical methods.

English is good, no major issues detected.
